# Aspirin Resistance Affects Medium-Term Recurrent Vascular Events after Cerebrovascular Incidents: A Three-Year Follow-up Study

**DOI:** 10.3390/brainsci10030179

**Published:** 2020-03-19

**Authors:** Adam Wiśniewski, Karolina Filipska, Joanna Sikora, Grzegorz Kozera

**Affiliations:** 1Department of Neurology, Faculty of Medicine, Nicolaus Copernicus University in Toruń, Collegium Medicum in Bydgoszcz, 85-094 Bydgoszcz, Poland; 2Department of Neurological and Neurosurgical Nursing, Faculty of Health Sciences, Nicolaus Copernicus University in Toruń, Collegium Medicum in Bydgoszcz, 85-821 Bydgoszcz, Poland; karolinafilipskakf@gmail.com; 3Experimental Biotechnology Research and Teaching Team, Department of Transplantology and General Surgery, Nicolaus Copernicus University in Toruń, Collegium Medicum in Bydgoszcz, 85-094 Bydgoszcz, Poland; joanna.sikora@cm.umk.pl; 4Medical Simulation Centre, Medical University of Gdańsk, Faculty of Medicine, 80-210 Gdańsk, Poland; gkozera1@wp.pl

**Keywords:** ischemic stroke, platelet reactivity, aspirin resistance, recurrent events, outcome

## Abstract

*Background*: The aim of this prospective, a three-year follow-up study, was to establish the role of high on-treatment platelet reactivity (HTPR) in predicting the recurrence of vascular events in patients after cerebrovascular incidents, particularly in the aspect of stroke etiology. *Methods*: The study included 101 subjects with non-embolic cerebral ischemia (69 patients with ischemic stroke and 32 patients with transient ischemic attack) treated with 150 mg of acetylsalicylic acid (aspirin) a day. The platelet reactivity was tested in the first 24 h after the onset of cerebral ischemia by impedance aggregometry. Recurrent vascular events, including recurrent ischemic stroke, transient ischemic attack, myocardial infarction, systemic embolism, or sudden death of vascular reason, were assessed 36 months after the onset of cerebral ischemia. *Results*: Recurrent vascular events occurred between 3 and 9 months after onset in 8.5% of all subjects; in the HTPR subgroup, recurrent vascular events occurred in 17.9%; in the normal on-treatment platelet reactivity (NTPR) subgroup, they occurred in 4.6%. We did not notice early or long-term recurrent events. Aspirin resistant subjects had a significantly higher risk of recurrent vascular events than did aspirin sensitive subjects (Odds ratio (OR) = 4.57, 95% Confidence interval (CI) 1.00–20.64; *p* = 0.0486). Cox proportional hazard models showed that large-vessel disease (Hazard ratio (HR) 12.04, 95% CI 2.43–59.72; *p* = 0.0023) and high on-treatment platelet reactivity (HR 4.28, 95% CI 1.02–17.93; *p* = 0.0465) were independent predictors of recurrent vascular events. Conclusion: Aspirin resistance in the acute phase of cerebral ischemia was associated with a higher risk of recurrent medium-term vascular events, coexisting with large-vessel etiology of stroke. Platelet function-guided personalized antiplatelet treatment should be considered for patients with recurrent strokes, especially when due to large-vessel disease.

## 1. Introduction

Disturbances in platelet function are associated with a higher risk of ischemic stroke, especially in the thrombotic mechanism [1]. The purpose of antiplatelet therapy used in secondary prevention after cerebrovascular incidents is to appropriately inhibit platelet activation and aggregation to reduce recurrent ischemic events [2]. According to the current European and American guidelines for stroke management, acetylsalicylic acid (ASA; aspirin) is still most often recommended as an antiplatelet agent, but its efficacy is variable [3]. Unfortunately, a certain group of patients, despite regular intake and the correct dose of ASA, show reduced platelet sensitivity, which leads to experiencing recurrent vascular events [4]. This phenomenon is termed high on-treatment platelet reactivity (HTPR), which equates to biochemical or laboratory aspirin resistance [5].

Previous literature reports demonstrated that aspirin resistance in stroke patients is associated with negative traits, including poor clinical condition, poor early and late prognosis, new ischemic changes, larger extent of brain lesions in neuroimaging, or a higher risk of death or recurrent vascular events [6,7]. In assessing the recurrence aspect, the focus was not on the time of occurrence of the event and usually the short-term follow-up period was taken into consideration. The impact of stroke etiology on the relationship between higher platelet activation and recurrent events was not analyzed. Therefore, the aim of this study was to estimate the time period of the most frequent occurrence of recurrent vascular events associated with aspirin resistance over the long-term (three-years) follow-up and to determine the significance of stroke etiology for the above relationships.

## 2. Materials and Methods

### 2.1. Study Population

This study was conducted from February 2016 to November 2019 in the Department of Neurology at the University Hospital No. 1 in Bydgoszcz. We prospectively enrolled 101 subjects with non-embolic cerebral ischemia (69 patients who met the American Heart Association/American Stroke Association (AHA/ASA) criteria for the diagnosis of ischemic stroke and 32 patients with transient ischemic attack (TIA)). Acetylsalicylic acid was administered at the time of admission at a dose of 150 mg to every patient. Due to the etiology of cerebral ischemia, we subdivided the subjects into a large-vessel disease group (with confirmation in the Doppler ultrasound of at least 50% stenosis of internal carotid artery on the side responsible for stroke symptoms) and a small-vessel disease group (with typical neuroimaging vascular lesions—subcortical and periventricular, leukoaraiosis features) [8]. 

The exclusion criteria included the inability to sign a consent form to participate in the study (moderate or severe aphasia, dementia, or alterations in consciousness), cardioembolic etiology of cerebral ischemia, oncological history, use of ASA before admission, chronic inflammatory processes affecting platelet function (chronic venous thrombosis or chronic lower limb ischemia), a history of cerebral ischemia in the previous two years, significant bleeding in the last two years, a platelet count <100 thousands/μL, a value of hematocrit <35%, or a level of hemoglobin <9 g/dL.

Sample size was calculated based on the prevalence of ischemic stroke (related to rigorous inclusion and exclusion criteria used in the manuscript) in the general population of province (Bydgoszcz, Poland) using an available sample size calculator. The recommended (estimated) sample size was 96 subjects with a confidence level of 95%.

The study was approved by the Bioethics Committee of Nicolaus Copernicus University in Torun at Collegium Medicum of Ludwik Rydygier in Bydgoszcz (KB No. 73/2016). All subjects read the study protocol and gave their informed consent form before enrollment.

### 2.2. Platelet Reactivity Research

The platelet function assay was performed from February 2016 to December 2016 by impedance aggregometry in the Laboratory of Experimental Biotechnology at the Collegium Medicum in Bydgoszcz using a Multiplate-Dynabyte multichannel platelet function analyzer (Roche Diagnostics, France). Whole blood samples were collected at a similar time (10:00–12:00) during the first 24 h after the onset of a cerebrovascular event. We used the acetylsalicylic acid platelet inhibition (ASPI) test, with a cofactor (arachidonic acid) as a platelet activator. We estimated that the platelet aggregation changes between two electrodes, that have been immersed in a whole blood sample. After the agonist was added, platelet aggregation increased, resulting in electrode deposition. Higher activation of platelets and their acummulation onto the electrodes led to a change in impedance (corresponds to electrical resistance) between them. The automatic apparatus then converted the obtained differences in impedance into telling graphs of platelet aggregation over time. The most important feature of the graphs was the area under the curve (AUC) value, based on which we assigned the subjects to specific groups. Aspirin resistance, which corresponds to high on-treatment platelet reactivity was defined as aggregation > 40 AUC. Subjects with AUC values of 40 and below were treated as sensitive to ASA, which corresponds to normal on-treatment platelet reactivity (NTPR). Impedance aggregometry was performed for all 101 subjects under the standard protocol [9,10]. Approximately 2.6 mL of blood was collected from the forearm veins in each subject into a Sarstedt r-hirudin-type tube. The blood was kept from 30 min. to max. 2 h. Then, 300 µL of whole blood was placed in a special chamber of Multiplate system with a magnetic stirrer, containing four consecutive measuring stations combined and linked to the device. After 3 min of incubation, 20 µL of arachidonic acid- platelet activator- was prepared and appended. The aggregation test duration was 6 min and the final result as AUC value was the average of the two measurements.

### 2.3. Follow-up Metrics

Patients were followed up with from February 2019 to November 2019 in the outpatient clinic during control monitoring visits or by telephone, in person, or with authorized family members with the staff at the Stroke Unit at the University Hospital No. 1 in Bydgoszcz. The full observation part lasted three years. Recurrent vascular events, including recurrent ischemic stroke, transient ischemic attack, myocardial infarction, systemic embolism, or sudden death due to a vascular reason, were assessed 36 months after the onset of cerebral ischemia.

### 2.4. Statistical Methodology

We used the Mann–Whitney U test to compare the continuous variables and Fisher’s exact test was used to compare categorical data. The continuous variables of non-normal distribution detected by the Shapiro–Wilk test were expressed as median and range. The Kaplan–Meier methodology was used to assess the cumulative survival of subjects compared to the platelet reactivity assays, and the log-rank test was used to evaluate the statistical differences between survival curves. We used the Cox regression hazards model to determine the risk factors of recurrent vascular events, and the Breslow estimator was used to create the survival curves. Logistic regression was used to compare the risk of recurrent vascular events between aspirin resistant and sensitive subjects. The significance level of *p* < 0.05 was considered statistically significant. All calculations were performed using STATISTICA software, version 13.1 (Dell Inc., Round Rock, TX, USA).

## 3. Results

### 3.1. Study Population

Based on the results of the platelet reactivity by impedance aggregometry, 101 patients were divided into HTPR and NTPR subgroups. The NTPR subgroup included 70 subjects and the HTPR included 31 subjects. The baseline characteristics of the two subgroups are listed in Table 1. Male gender, smoking, and large-vessel disease etiology of cerebral ischemia were associated with the HTPR subgroup compared with the patients in the NTPR subgroup. A total of seven subjects (6.9%) were lost to follow up (three in the HTPR subgroup and four in the NTPR subgroup). Two patients died during the study as a result of advanced cancer disease and the connection to five other patients was lost.

### 3.2. Cinical Outcomes

A total of eight subjects of all 94 patients (8.5%) experienced recurrent vascular events during a three-year follow-up. In the HTPR subgroup recurrent events occurred in five subjects of total 28 followed-up subjects (17,9%): three recurrent ischemic strokes, one transient ischemic attack, and one myocardial infarction. In the NTPR subgroup, recurrent events occurred in three subjects of the total 66 followed-up subjects (4.6%): one recurrent ischemic stroke, one transient ischemic attack, and one sudden death of vascular reason. Compared with the NTPR subgroup, the HTPR subgroup exhibited a higher risk of recurrent vascular events (log-rank test *p* = 0.0323). Vascular event-free survival curves in two analyzed subgroups are presented in Figure 1. The logistic regression showed that aspirin resistant subjects have a significantly higher risk of recurrent vascular events than aspirin sensitive subjects (Odds ratio (OR) = 4.57, 95% confidence interval (CI) 1.00–20.64; *p* = 0.0486).

### 3.3. Predictors of Recurrent Vascular Events

Based on the presence of recurrent vascular events we subdivided the 94 patients who had completed the 36 months of follow-up into recurrent and non-recurrent vascular event subgroups. The large-vessel disease etiology of cerebral ischemia (*p* = 0.0014), aspirin resistance (*p* = 0.0484), and high platelet reactivity by impedance aggregometry (*p* = 0.0339) were significantly associated with the recurrent vascular events subgroup compared with subjects with non-recurrent vascular events (Table 2).

The univariate Cox proportional hazard models show that large-vessel disease (Hazard ratio (HR) 12.04, 95% CI 2.43–59.72; *p* = 0.0023) and high on-treatment platelet reactivity (HR 4.28, 95% CI 1.02–17.93; *p* = 0.0465) were significant predictors of recurrent vascular events (Table 3).

Breslow estimators showing the role of two above predictors for probability of recurrent vascular event-free survival are presented in Figure 2 and Figure 3. The multivariate Cox regression adjusted for age, sex, and type of cerebral ischemia showed that high on-treatment platelet reactivity (HR 5.31, 95% CI 1.2–23.45; *p* = 0.0275) and large-vessel disease (HR 39.64, 95% CI 2.64–595.01; *p* = 0.0078) are independently associated with recurrent vascular events.

## 4. Discussion

The present study demonstrated an increased rate of recurrent vascular events in patients with cerebral ischemia presenting high on-treatment platelet reactivity. The first novelty of our study was a long time of observation—we followed up with patients 36 months after the ischemic event onset. The second novelty was the association with medium-term recurrent events, but not with early or long-term events. 

In our study, all recurrent events occurred from 3 to 9 months after the onset. We did not observe early (up to 60 days) or any late incidents in the second and third year after onset. Most of the recent results were estimated during the 3–12 month period of follow-up [11,12,13], with only one up to 24 months [14]. Rao et al. demonstrated only early events within the first 90 days after onset, but they enrolled only the Chinese population with minor stroke or high risk TIA and performed different platelet function method assays (thromboelatography (TEG)) and time-point (seven days after stroke) [11]. 

Yi et al. [14] followed up with subjects for up to 24 months and Zhang et al. and Jing et al. [12,13] up to 12 months; however, they did not analyze and focus in their study on the time-point of the events, so there is no data to compare. Kim et al. [15] demonstrated the early correlation with new ischemic lesions in the brain within the first five days after cerebral ischemia, but not with clinical deterioration. We had a hypothesis that in our group that some cases of silent recurrent stroke could occur in the first days after stroke onset, but that they were not noticed as subclinical events or that the symptoms were underestimated by the subjects and therefore they were not reported in our study. On the other hand, the long-term recurrent events seem not to be associated with aspirin resistance. However, further trials are needed to validate our findings.

The association between aspirin resistance and a higher risk of vascular events has been reported in previous studies. Most of them also analyzed multiple vascular events, such as stroke/TIA, myocardial infarction, sudden deaths from vascular reasons [11,14], only cardiovascular mortality [13], only stroke/TIA [7], or both stroke/TIA and deaths [12]. Most of these studies were performed on the Chinese population, and different platelet function methods and time points or doses of ASA were used to this purpose. Despite these differences, we found a similar prevalence of aspirin resistance (Rao et al. −22,7%, Yi X. et al. −20,4%, Zhang et al. −20.1%, Fiolaki et al. −23%) and confirmed that aspirin resistance is a predictor of recurrent events. 

Patients with HTPR exhibited a higher risk for recurrent vascular events compared with patients with NTPR (17,9% vs. 4,6%). We noticed similar results as Rao et al. (HTPR vs. NTPR; 18.9% vs. 5.8% *p* = 0.006) and lower levels than Yi et al. (HTPR vs. NTPR; 31.3% vs. 12.4% *p* < 0.001) and Zhang et al. (HTPR vs. NTPR; 46.6% vs. 17.5%; *p* < 0.001), but found the same tendency. Zhang et al. enrolled all types of stroke, including cardioembolic. This could explain the higher rates of recurrent events as opposed to ours. A multicenter meta-analysis verified that aspirin-resistant subjects have a higher risk (relative risk (RR) = 1.81) of recurrent cerebrovascular events [7]. Consistent with those results, we demonstrated a significant role of aspirin resistance as an independent predictor of recurrent multiple vascular events.

The data from the literature demonstrated that the highest rate of recurrent events was noticed during the first year after ischemic stroke/TIA at the rate of 10–12%, and it decreased to 5–8% in every year after onset [16]. The next stroke/TIA, particularly with the same etiology, is seen the most often from all recurrent vascular events, and it is about 75% [1]. In our study, we obtained the same result in the aspect of recurrent cerebral ischemia frequency and we proved that cerebrovascular incidents are the most common. Similar results (77,7%) were presented in another study [14]. However, our findings showed that the prevalence of all recurrent vascular events was on the lower level in the first year (8,5%) and significantly different in the next 24 months, where we did not notice any late events. Likely, this was associated with the exclusion of patients with an embolic etiology of stroke/TIA where oral anticoagulants are the standard therapy and the role of platelet reactivity is decreased. The cardioembolic etiology of stroke is estimated as the most associated with recurrent vascular events [12].

Another novelty presented in this study is underlining the coexistence of aspirin resistance with the large-vessel disease etiology of cerebrovascular incidents and emphasizing the role of this etiology of stroke with a higher risk of recurrent events. In our previous study, we confirmed the role of large artery atherosclerosis for assessment of a significant relationship between platelet reactivity and the extent of brain ischemic lesions [6]. In this paper, we found that large-vessel disease is associated with high on-treatment platelet reactivity, and aspirin resistance and large vessel disease are independently associated with a higher rate of recurrent vascular events. Other authors did not raise the role of etiology for correlations between aspirin resistance and the recurrence of vascular events. Despite that they most often included subjects with large-vessel and small-vessel disease, no correlation with the recurrence of vascular events was reported in the aspect of etiology. We found some data that the large-vessel disease etiologic subtype of stroke is associated with a higher risk of recurrent events [17,18], but the authors did not notice and instead emphasized the coexisting role of high on-treatment platelet reactivity in this field.

On the basis of the results of our study, we suppose that in many cases of patients with carotid artery stenosis, the regular treatment with ASA at a dose of 150 mg is insufficient. The AHA/ASA and European guidelines for the management of acute stroke did not differentiate the antiplatelet treatment based on the etiology of stroke, and they recommend a dose of 75–320 mg ASA as the drug of first choice regardless of whether the reason of the stroke is small- or large-vessel disease [19,20]. Dual antiplatelet treatment during the first 21 days after a minor stroke or high risk TIA is recommended without any connection with the etiology [21]. 

Taking into account the results of this study, our previous results, and the data reported by other authors, the same treatment should be considered among stroke patients caused by large artery atherosclerosis with recognized high on-treatment platelet reactivity. We estimate that, in this etiologic subtype of stroke, more intensive secondary prevention might result in a sufficient reduction in recurrent vascular events. On the other hand, the results of our study support the need for urgent carotid imaging and prompt aggressive medical management, include invasive procedures, such as stenting or endarterectomy. We posit that the routine assessment of platelet reactivity and the introduction of the platelet function-guided individualized treatment of stroke could be beneficial for large-vessel disease subjects.

We fully agree with the authors who customized antiplatelet therapy based on platelet function assays and their results that showed lower rates of recurrent events after platelet reactivity-guided modification of therapy [22,23]. In addition, there have been reports about dual antiplatelet therapy (ASA plus clopidogrel) that could be effective and beneficial for the large artery atherosclerosis etiologic subtype of stroke [24,25] and that dual antiplatelet therapy is associated with a lower risk of HTPR [7]. Considering the results obtained in our study, this provides further support for updating the stroke guidelines and recommendations. More research is needed to determine personalized antiplatelet therapy in strokes due to large-vessel disease coexisting with high on-treatment platelet reactivity; however, platelet function-guided management of strokes should be considered.

Several limitations of our study should be considered. The sample size was small. Our conclusions were based only on one-time point results of platelet function using a single, poorly standardized assay. It is possible that in individual cases, subjects with undetected cardioembolic etiology of stroke were enrolled. We excluded stroke subjects with severe neurological conditions who were unable to sign informed consent, and this limited the cross-section of stroke patients in the research. Authors are aware that some significant odds ratios and hazard ratios reported in our paper had wide confidence intervals and several main findings showed weak correlation and borderline statistical significance. We consider small sample size, wide ranges of presented data, variability in platelet function measurements, high confidence level and probably collinearity of variables as main reasons affecting the above- mentioned limitations in statistical calculations. Nevertheless, every effort has been made to standardize and to unify the measurements to improve the significance level of the results obtained in this study.

## 5. Conclusions

Aspirin resistance affects recurrent vascular events in patients after the cerebrovascular incident. In a three-year follow-up period, there was a clear trend toward medium-term events. We highlighted the coexistence of high on-treatment platelet reactivity with the large-vessel disease etiology of stroke, as these are independently associated with the risk of major recurrent vascular events. Platelet function testing and platelet reactivity-guided individualized antiplatelet treatment should be considered for patients with recurrent stroke, particularly when due to large-vessel disease.

## Figures and Tables

**Figure 1 brainsci-10-00179-f001:**
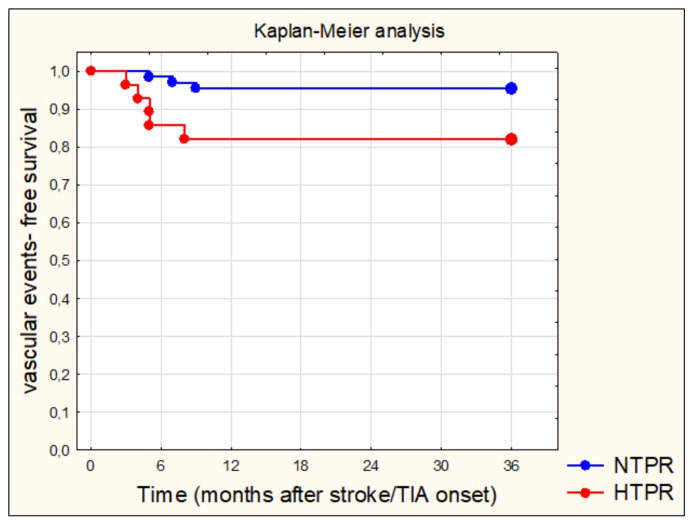
Curves of the probability of vascular event-free survival comparing normal on-treatment platelet reactivity (NTPR) subjects—blue line; and high on-treatment platelet reactivity (HPRT) subjects—red line. Significant lower probability of vascular events- free survival over time has been demonstrated in HTPR subjects.

**Figure 2 brainsci-10-00179-f002:**
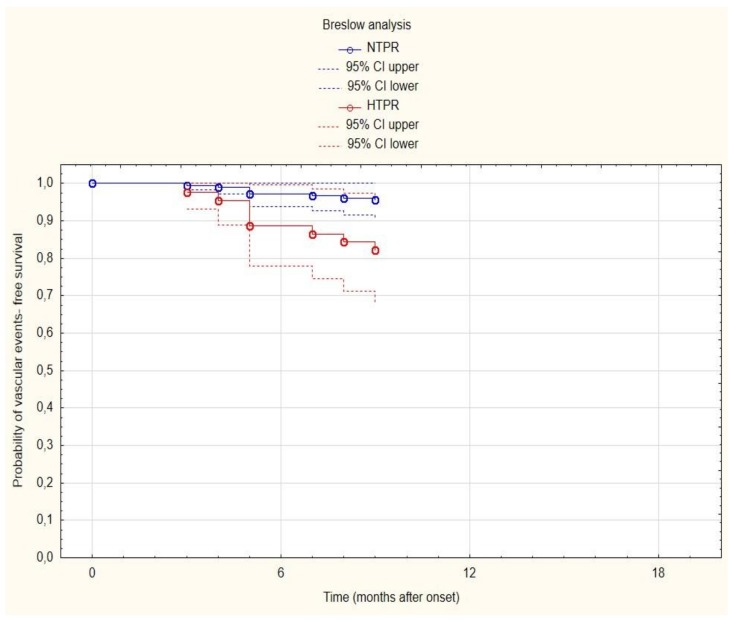
Curves of the probability of vascular event-free survival depending on time (for transparency, reduced to 18 months) in normal on-treatment platelet reactivity (NTPR) subjects—blue line; and high on-treatment platelet reactivity (HPRT) subjects—red line.

**Figure 3 brainsci-10-00179-f003:**
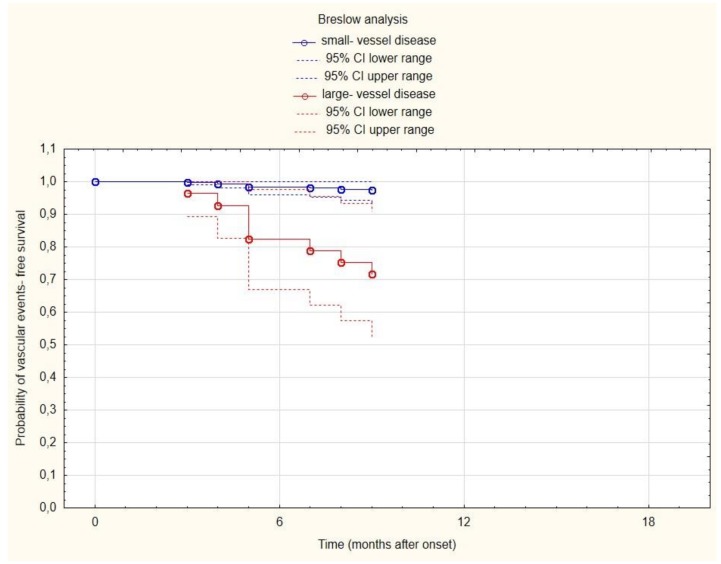
Curves of the probability of vascular event-free survival depending on time (for transparency, reduced to 18 months) in small-vessel disease subjects—blue line; and large-vessel disease subjects—red line.

**Table 1 brainsci-10-00179-t001:** Baseline characteristics of the high on-treatment platelet reactivity (HTPR) and normal on-treatment platelet reactivity (NTPR) subgroups.

Parameter	HTPR	NTPR	*p*-Value
	*N* = 31	*N* = 70	
Age median in years (range) *	65 (60–74)	69.5 (64–80)	0.066
Male N (%) **	20 (64.5%)	28 (40%)	**0.0195**
Ischemic stroke N (%) **	22 (71%)	47 (67.1%)	0.4451
Large- vessel disease N (%) **	12 (38.7%)	9 (12.9%)	**0.0045**
BMI median (range) *	29.23 (24.66–32.88)	27.55 (24.77– 29.74)	0.3334
NIHSS at admission median (range) *	5 (3.5–7.5)	5 (5–9)	0.3693
mRS at admission median (range) *	2 (1–3)	2 (1–3)	0.8225
Recurrent vascular events N (%) **	5 (17.9%)	3 (4.6%)	**0.0381**
Hypertension N (%) **	25 (80.6%)	62 (88.5%)	0.2228
Diabetes N (%) **	9 (29.5%)	27 (38.6%)	0.2442
Hyperlipidemia N (%) **	14 (45.1%)	31 (44.3%)	0.5526
Smoking N (%) **	16 (51.6%)	14 (20%)	**0.0018**
Ischemic heart disease N (%) **	2 (6.5%)	11 (15.7%)	0.1695
CRP (mg/l) median (range) *	3.53 (1.89-7.18)	3.49 (1.58-7.83)	0.808
HBA1c (%) median (range) *	5.7 (5.4-6.7)	5.75 (5.4-6.4)	0.9941
Homocystein (umol/l) median (range) *	9.75 (7.34-12.92)	10.53 (8.79-13)	0.1418
Fibrinogen (mg/dl) median (range) *	290 (256-365)	310 (258-372)	0.6719

* Mann–Whitney U test, ** Fisher’s exact test; BMI, body mass index; CI, confidence interval; NIHSS, National Institute of Health Stroke Scale; mRS, modified Rankin scale; CRP, C-reactive protein; AUC, area under the curve; HBA1c, hemoglobin A1c. Bold font identifies statistically significant relationships.

**Table 2 brainsci-10-00179-t002:** The baseline characteristics of groups with recurrent events and non-recurrent events.

Parameter	Recurrent Events Group *N* = 8	Non-recurrent Events Group *N* = 86	*p*-Value
Age in years median (range) *	74.5 (64.5–85)	71 (69–80)	0.2690
Male N (%) **	5 (62.5%)	39 (45%)	0.2877
Ischemic stroke N (%) **	6 (75%)	57 (66%)	0.4727
Large-vessel disease N (%) **	6 (75%)	15 (17.4%)	**0.0014**
BMI median (range) *	29.05 (28.34–29.74)	27.28 (24.62–30.99)	0.5527
NIHSS (points) at admission median (range) *	5.5 (5–6)	5 (4–9)	0.7454
mRS (points) at admission median (range) *	4 (3–4)	4 (3–4)	0.6961
Aspirin resistance N (%) **	5 (62.5%)	23 (26.7%)	**0.0484**
Platelet reactivity (AUC) median (range) *	64 (23.5–83)	26 (18–41)	**0.0339**
Hypertension N (%) **	5 (62.5%)	76 (88.4%)	0.0076
Diabetes N (%) **	3 (37.5%)	30 (34.9%)	0.5808
Hyperlipidemia N (%) **	4 (50%)	40 (46.5%)	0.5686
Smoking N (%) **	3 (37.5%)	26 (30.2%)	0.4720
Ischemic heart disease N (%) **	0 ( 0%)	12 (13.9%)	0.3202
CRP (mg/l) median (range) *	7.08 (4.15–9.96)	3.29 (1.58–7.17)	0.0644
HBA1c (%) median (range) *	5.9 (5.75–7.0)	5.70 (5.4–6.4)	0.2042
Homocystein (umol/l) median (range) *	8.8 (6.51–12.68)	10.18 (8.37–12.92)	0.3533
Fibrinogen (mg/dl) median (range) *	310 (281–387)	308 (248–365)	0.3675

* Mann–Whitney U-test, ** Fisher’s exact test. BMI, body mass index; NIHSS, National Institute of Health Stroke Scale; mRS, modified Rankin sale; CRP, C-reactive protein; AUC, area under the curve; HBA1c, hemoglobin A1c. Bold font identifies statistically significant relationships.

**Table 3 brainsci-10-00179-t003:** Predictors of recurrent vascular events in study population by Cox proportional hazard models.

Parameter	HR	95% CI	*p*-Value
Age in years median (range) *	1.04	0.98–1.11	0.2209
Male N (%) **	2.01	0.48–8.40	0.3401
Ischemic stroke N (%) **	0.68	0.14–3.37	0.6376
Large-vessel disease N (%) **	**12.04**	**2.43–59.72**	**0.0023**
BMI median (range) *	1.00	0.85–1.16	0.9899
NIHSS (points) at admission median (range) *	0.89	0.69–1.16	0.4174
mRS (points) at admission median (range) *	1.29	0.53–3.12	0.5693
Aspirin resistance N (%) **	**4.28**	**1.02–17.92**	**0.0467**
Platelet reactivity (AUC) median (range) *	**1.03**	**1.00–1.05**	**0.0149**
Hypertension N (%) **	4.16	0.99–17.43	0.0512
Diabetes N (%) **	0.93	0.22–3.90	0.9253
Hyperlipidemia N (%) **	0.86	0.21–3.41	0.8241
Smoking N (%) **	0.73	0.17–3.07	0.6698
Ischemic heart disease N (%) **	1.02	0.99–1.04	0.9933
CRP (mg/l) median (range) *	1.04	0.95–1.14	0.4392
HBA1c (%) median (range) *	1.32	0.81–2.14	0.2594
Homocystein (umol/l) median (range) *	1.01	0.89–1.13	0.9078
Fibrinogen (mg/dl) median (range) *	1.00	0.99–1.01	0.3140

* Mann–Whitney U-test, ** Fisher’s exact test, HR, Hazard ratio; BMI, body mass index; CI, confidence interval; NIHSS, National Institute of Health Stroke Scale; mRS, modified Rankin sale; CRP, C-reactive protein; AUC, area under the curve; HBA1c, hemoglobin A1c. Bold font identifies statistically significant relationships.

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
