# Peer review of "Aspirin Resistance Affects Medium-Term Recurrent Vascular Events after Cerebrovascular Incidents: A Three-Year Follow-up Study"

_brainsci, 2020, doi:10.3390/brainsci10030179_

Round 1

Reviewer 1 Report

This paper is of interest to stroke research and presents some interesting data. There is lack of information regarding the impedance methodology and the statistical analysis is generally appropriate although the authors to my knowledge do not explain how the sample size (if at all) was determined. .

My main concern is that  several of the significant OR and HR reported  have very wide confidence intervals. These include recurrent aspirin resistant with a significantly higher risk of recurrent vascular events having an OR of 4.57 but a confidence interval of 1.0 to 20.64 p=0.0465. The authors should comment on the fact that this and other OR/HRs include unity and suggests that this group is not at higher risk. The authors comment that this is only a "moderate" sized sample as a limitation but these wide confidence intervals should be discussed in greater detail and may be associated with large Standard Deviations that can be masked in their data by reporting medians and ranges. In addition did they check their data for colinearity another possible cause of wide confidence intervals. 

Many of the Tables lack clarity and poorly synchronised. The Kaplan Meier curve labeling is very unclear and are of poor legibity.

Reviewer 2 Report

The article is interesting and can be published with some minor corrections.
In some parts, make the speech more fluid. Use greater lexical accuracy.
Tables 1 and 2 need to be organized more clearly.

Round 2

Reviewer 1 Report

My comments refer to the re-submitted amended paper.

The authors have dealt with my comments and criticisms in an appropriate manner. I know feel that the paper is acceptable.